# Structural plasticity of SARS-CoV-2 3CL M^pro active site cavity revealed by room temperature X-ray crystallography

Daniel W. Kneller [1], Gwyndalyn Phillips [1], Hugh M. O'Neill [1], Robert Jedrzejczak[2,3], Lucy Stols[2], Paul Langan[1], Andrzej Joachimiak[2,3,4], Leighton Coates [1✉] & Andrey Kovalevsky [1✉]

The COVID-19 disease caused by the SARS-CoV-2 coronavirus has become a pandemic health crisis. An attractive target for antiviral inhibitors is the main protease 3CL M^pro due to its essential role in processing the polyproteins translated from viral RNA. Here we report the room temperature X-ray structure of unliganded SARS-CoV-2 3CL M^pro, revealing the ligand-free structure of the active site and the conformation of the catalytic site cavity at near-physiological temperature. Comparison with previously reported low-temperature ligand-free and inhibitor-bound structures suggest that the room temperature structure may provide more relevant information at physiological temperatures for aiding in molecular docking studies.

---

[1] Neutron Scattering Division, Oak Ridge National Laboratory, 1 Bethel Valley Road, Oak Ridge, TN 37831, USA. [2] Center for Structural Genomics of Infectious Diseases, Consortium for Advanced Science and Engineering, University of Chicago, Chicago, IL 60667, USA. [3] Structural Biology Center, X-ray Science Division, Argonne National Laboratory, Argonne, IL 60439, USA. [4] Department of Biochemistry and Molecular Biology, University of Chicago, Chicago, IL 60367, USA. ✉email: coatesl@ornl.gov; kovalevskyay@ornl.gov

A new coronavirus named severe acute respiratory syndrome coronavirus (SARS) 2, or SARS-CoV-2, caused a world pandemic disease called COVID-19[1–4]. A significant research push is now underway to repurpose existing drugs and to design new therapeutic agents targeting various components of the virus[5]. The viral single-stranded RNA genome is 82% identical to the earlier SARS coronavirus (SARS-CoV) with some viral proteins being more than 90% homologous to SARS-CoV[6]. SARS-CoV-2, similar to many other single-stranded RNA viruses, employs a chymotrypsin-like protease (3CL main protease, or 3CL M$^{pro}$) to enable the production of non-structural proteins essential for viral replication[7–9].

3CL M$^{pro}$ cleaves two large overlapping polyproteins pp1a and pp1ab at least 11 conserved sites, including its own N-terminal and C-terminal autoprocessing sites. The enzyme has a recognition sequence of Leu-Gln↓Ser-Ala-Gly, where ↓ marks the cleavage site, but shows sequence promiscuity. The absolute dependence of the virus on the correct function of this protease, together with the absence of a homologous human protease, makes 3CL M$^{pro}$ an attractive, albeit difficult, target for the design of specific protease inhibitors[10]. Unfortunately, to date, no protease inhibitors targeting SARS-CoV 3CL M$^{pro}$ have been FDA-approved, despite significant research effort during the past fifteen years[11–17].

The 3CL M$^{pro}$ structure is composed of three domains[18,19]. Domains I (residues 8–101) and II (residues 102–184) are composed of antiparallel β-barrel structures and are the catalytic domains. Domain III (residues 201–303) is composed of five α-helices and is responsible for the enzyme dimerization. Based on studies of SARS-CoV 3CL M$^{pro}$ this helical domain plays an essential role in the protease function as the monomeric enzyme is not catalytically active[20–24]. Thus, 3CL M$^{pro}$ forms a functional dimer through intermolecular interactions, mainly between the helical domains (Fig. 1a).

3CL M$^{pro}$ is uniquely diversified to have an unconventional Cys catalytic residue. Unlike other chymotrypsin-like enzymes and many Ser (or Cys) hydrolases, it has a catalytic Cys-His dyad instead of a canonical Ser(Cys)-His-Asp(Glu) triad[8]. The catalytic residues Cys145 and His41 in 3CL M$^{pro}$ are buried in an active site cavity located on the surface of the protein. This cavity can accommodate four substrate residues in positions P1′ through P4, and it is flanked by residues from both domains I and II (Fig. 1b).

We present here atomic details pertinent to the function and inhibitor binding to SARS-CoV-2 3CL M$^{pro}$. To gain these insights we determined a room temperature (293 K) X-ray structure of the enzyme to 2.30 Å resolution that provides a proper and accurate physiologically relevant template for structure-assisted drug design and molecular simulations.

## Results

**Atomic details of 3CL Mpro active site at room temperature.** We grew large crystals (Supplementary Fig. 1) that could be used on a home source to ensure minimal radiation damage. In our structure of ligand-free 3CL M$^{pro}$, the catalytic Cys145 Sγ is 3.8 Å from His41 Nε2, which appears to be too long for the formation of a hydrogen bond (Fig. 2). This is not surprising, taking into account the experimental pK$_a$ values of 8.0 ± 0.3 for Cys145 and 6.3 ± 0.1 for His41 measured previously for the SARS 3CL M$^{pro}$ that shares 96% homology with the SARS-CoV-2 enzyme[25,26] and the poor hydrogen bonding properties of thiols. Thus, in our crystallization conditions (see "Crystallization" section in "Methods") at the pH in the crystallization drop of 7.0, both catalytic residues are expected to be uncharged adopting the ligand-free enzyme's state before substrate or inhibitor binding.

In this ligand-free state, the thiol of Cys145 is protonated, and the imidazole of His41 is neutral. The catalytic dyad would be activated by a proton transfer from Cys145 to His41 possibly triggered by substrate binding or occurring in a transition state during the attack by the sulfur on the carbonyl carbon atom of the scissile peptide bond. Conversely, His41 makes a strong hydrogen bond with a water molecule (suggestively named H$_2$O$_{cat}$), which in turn is stabilized through hydrogen bonds of 2.9 and 3.0 Å with the side chains of Asp187 and His164, respectively. The position of Asp187 is further stabilized through a salt-bridge with the nearby residue Arg40.

H$_2$O$_{cat}$ is involved in a complex network of interactions, mediating polar contacts between the catalytic His41, a conserved His164, and a conserved Asp187 located in the domain II–III junction. It is not unreasonable to suggest that this water may play a role of the third catalytic residue, completing the non-canonical catalytic triad in 3CL M$^{pro}$ and acting to stabilize the positive charge on His41 by mediating its electrostatic interaction with the negatively charged Asp187 during catalysis. We note that in some X-ray structures of the ligand-free 3CL M$^{pro}$ from SARS-CoV-2 (e.g., PDB ID 6M03) obtained at 100 K, this potentially crucial water molecule is absent.

Unsurprisingly, a significant number of reports have now appeared in which 100 K X-ray structures of the ligand-free 3CL M$^{pro}$ have been used for molecular docking simulations of

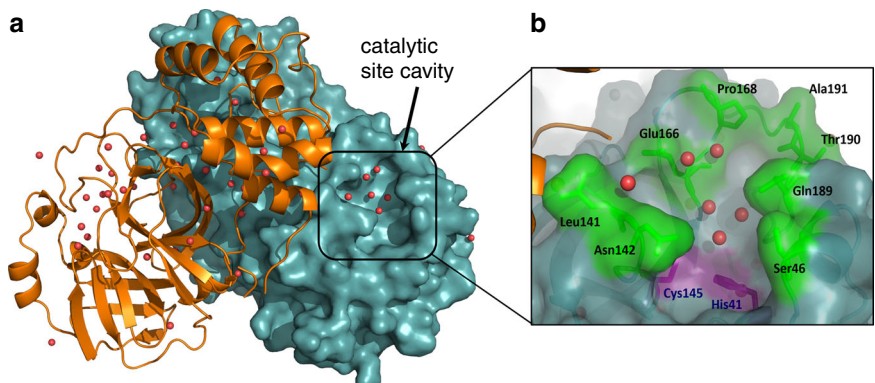

**Fig. 1 The three-dimensional structure of 3CL M$^{pro}$ from SARS-CoV-2. a** One monomer of the dimer is shown as an orange cartoon, while the other monomer is shown as a teal surface with the catalytic site cavity highlighted with water molecules shown as red spheres. **b** A closeup view of the catalytic site cavity in which the catalytic residues (Cys145 and His41) are highlighted in purple with the residues that flank the cavity highlighted in green with water molecules shown as red spheres.

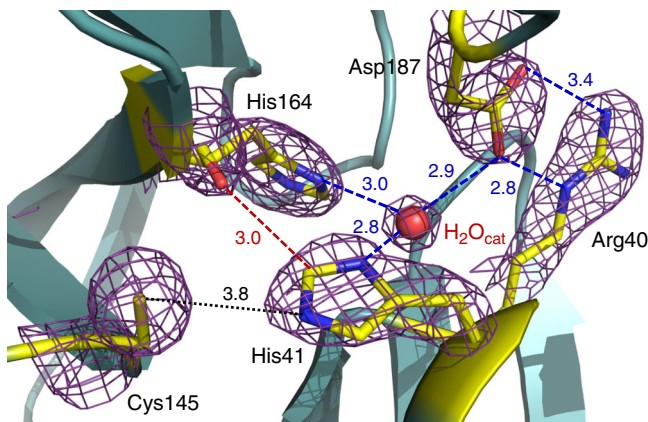

**Fig. 2 The catalytic site of 3CL M$^{pro}$ from SARS-CoV-2.** Hydrogen bonds are shown as blue dashed lines; the distance between Cys145 and His41 is shown as a black dotted line, the dashed red line indicates a strong C–H...O bond. The 2F$_O$ – F$_C$ electron density map contoured at 1.6 σ level is shown as a violet mesh. All distances are given in Ångstroms.

various small molecules, including many of the therapeutics approved to treat other diseases. Using a least-squares fit in the Coot molecular graphics program[27] we superimposed our room temperature structure of 3CL M$^{pro}$ with one obtained at 100 K (PDB ID 6Y2E)[18]. Overall, structures are similar with an R.M.S. D. for C$_\alpha$ atoms of 0.32 Å (Fig. 3a). The conformation of residues 192–198 differs between the room temperature and 100 K structures (Fig. 3b). The peptide bond of Ala194 is flipped in the room temperature structure pointing inwards into the P5 inhibitor binding pocket where it adopts a conformation similar to that seen in 3CL M$^{pro}$ in complex with inhibitor N3 (PDB ID 6LU7)[19]. Residues Thr196 and Asp197 also differ significantly in their conformations between the room temperature and 100 K structures. The backbone carbonyl oxygen atom of Thr 196 differs in position by 1.3 Å, the CG atoms of Asp197 are separated by 1.9 Å, and the position of backbone carbonyl oxygen atoms of Asp 197 differs in position by 2.6 Å. The conformations observed in the ligand-free enzyme at room temperature may be more relevant for the screening of possible drug candidates.

**Ligand binding induces active site conformational changes.** It is also instructive to compare our room temperature structure of the protease with the structure of an inhibitor-bound complex. For this comparison, we chose the complex with a structurally long peptidomimetic inhibitor N3[19] because it has substituents spanning all substrate binding subsites, including substituents at positions P4 and P5, thus closely resembling an actual substrate. Figure 4 shows the superposition of the two structures. The structural comparison reveals significant structural plasticity of the enzyme in the vicinity of the active site. To accommodate the inhibitor several secondary-structure elements move by more than 1 Å away from their positions in the room temperature structure of the ligand-free form. Such conformational changes can be characterized as induced fit due to ligand binding.

On ligand binding, the small helix near P2 group containing residues 46–50 and the β-hairpin loop near P3–P4 substituents with residues 166–170 shift apart by 2.4 Å, whereas the P5 loop spanning residues 190–194 moves closer to the P3–P4 loop. Two methionines, Met49 and Met165, avoid clashing with the inhibitor's leucine at position P2 by altering their side-chain conformations in the structure of the complex. Further, the change in Met49 conformation cascades to changes in the side

chain positions of Ser46 and Leu50. More dramatic conformational changes due to inhibitor binding occur at the enzyme's C-termini. Unexpectedly, the C-terminal tail consisting of residues Ser301 through Gln306 swings 180° from its position in the room temperature ligand-free structure and is situated above the helical domain in the N3 inhibitor-bound form (Supplementary Fig. 2).

The drastic flip in the C-terminal loop conformation eliminates several hydrogen bonds made as part of the dimer interface in the ligand-free form, which may destabilize the dimer in the inhibitor-bound form to a certain degree. To assess the flexibility of these enzyme regions, we performed a 1 μs molecular dynamics (MD) simulation of the ligand-free 3CL M$^{pro}$. As shown in Supplementary Fig. 3, in our MD simulation the same regions, including the P2 helix (residues 45–50), the P5 loop (residues 190–194), and the C-terminal tail are the most dynamic, showing the largest root-mean square fluctuations (RMSF). Therefore, these structural regions are quite malleable, possibly able to accommodate various chemical groups at the P2–P5 sites of inhibitors.

The conformational flexibility of the enzyme active site detected by comparisons between the room temperature ligand-free structure reported here with the low-temperature ligand-free and inhibitor-bound structures previously reported leads us to suggest that room-temperature structure of the 3CL M$^{pro}$ ligand-free form may be the more physiologically relevant structure for performing molecular docking studies to estimate drug binding and enable drug design.

## Methods

**General information.** Protein purification supplies were purchased from GE Healthcare (Piscataway, New Jersey, USA). Crystallization reagents were purchased from Hampton Research (Aliso Viejo, California, USA).

**Cloning of M$^{pro}$ gene to MBP self-cleavable fusion.** The 3CL M$^{pro}$ (Nsp5 M$^{pro}$) from SARS CoV-2 was cloned similarly to SARS-CoV M$^{pro}$ (PMID: 17189639) with the exception that upstream protein used was MBP instead of original GST. The gene for 3CL M$^{pro}$ SARS-CoV-2 optimized for *E. coli* expression was synthesized (Supplementary Table 1) and cloned directly into pET15b vector (Bio Basic) and named M$^{pro}$-pET15b. The M$^{pro}$ gene was amplified from M$^{pro}$-pET15b using the following primers: 5′-gggttggaagttttgagcgctgttctgcagtctggtttccgt and 5′-gtgatggtgatgatgcggaccctggaaggtaacaccagagcactga followed by treatment with T4 polymerase in the presence of dGTP. The vector for inserting M$^{pro}$: MBP-TEV-His$_7$ fragment from pMHTDelta238 (PMID: 17543538) was cloned to vector pMCSG81 and named pMCSG81-Delta238. For cloning of M$^{pro}$, pMCSG81-Delta238 was amplified with the following primers: 5′-catcatcaccatcaccattga-gatccggctgctand and 5′-caaaacttccaacccggccaccgtcgccgttaat. The PCR product was purified and T4 treated in the presence of dCTP. Resulting T4 treated fragments were mixed and transformed into BL21-Gold(DE3) cells (Agilent, Santa Clara, CA) and selected against ampicillin. Plasmid from a single colony was purified and sequenced. The plasmid name was designated as pCSGID-Mpro. In this expression system at the N-terminus, the construct is flanked by the maltose binding protein followed by the 3CL M$^{pro}$ autocleavage site SAVLQ↓SGFRK (arrow indicates the cleavage site) corresponding to the cleavage between NSP4 and NSP5 in the viral polyprotein. At the C-terminus, the construct codes for the human rhinovirus 3C PreScission protease cleavage site (SGVTFQ↓GP) connected to a His$_6$ tag. The authentic N-terminus is generated by 3CL M$^{pro}$ autoprocessing during expression, whereas the authentic C-terminus is generated by the treatment with PreScission protease, similar to the published methodology[18].

**Protein expression and purification.** Expression of 3CL M$^{pro}$ using Luria-Bertani, supplemented with 1 g l$^{-1}$ glucose, was performed in *E. coli* (BL21-DE3) cells using carbenicillin antibiotic (150 mg l$^{-1}$ of culture). The cells were grown to an OD$_{600}$ of 0.8 at 37 °C before induction with the addition of 0.2 mM isopropyl-D-thioga-lactoside. The temperature was then dropped to 18 °C and 3CL M$^{pro}$ was over-expressed for 18 h. The harvested cells were resuspended in the lysis buffer containing 20 mM TRIS pH = 8, 40 mM imidazole, 150 mM NaCl and 1 mM TCEP. After cells were lysed by sonication the insoluble fraction was removed by centrifugation at 30,000 × g for 30 min; the supernatant was then loaded onto a HisTrap FF column. 3CL M$^{pro}$ was eluted using a linear gradient of buffer containing 20 mM TRIS pH = 8, 500 mM imidazole, 150 mM NaCl and 1 mM TCEP. The fractions containing the protease were then pooled, and PreScission protease containing a His$_6$ tag (Sigma-Aldrich, St. Louis, MO) was added at a 500:1 molar ratio. The mixture was then dialyzed against a solution containing 20 mM TRIS

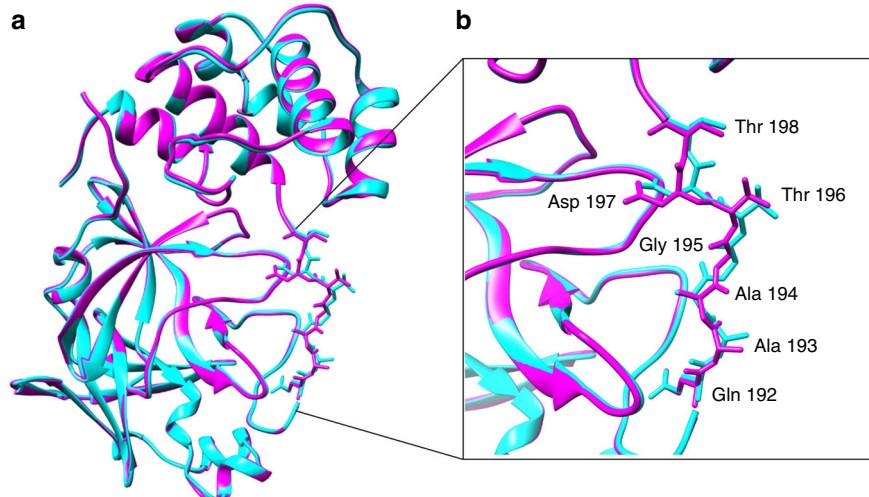

**Fig. 3 Comparison of room-temperature and low-temperature structures. a** A superposition of our room temperature ligand-free structure of 3CL M$^{pro}$ (magenta) with the ligand-free structure of 3CL M$^{pro}$ (PDB ID 6Y2E) obtained at 100 K (cyan). **b** Residues 192–198 in the P5 binding pocket differ in conformation between the room temperature and 100 K structures.

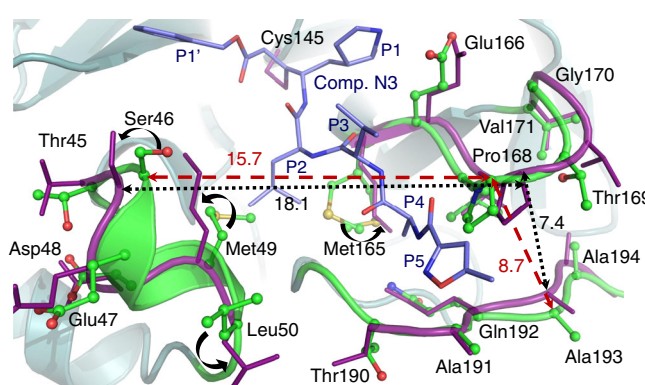

**Fig. 4 Comparison of the active site geometries in the ligand-free and ligand-bound structures.** Superposition of the room temperature ligand-free structure of 3CL M$^{pro}$ (green carbon atoms) with the structure of 3CL M$^{pro}$ in complex with inhibitor N3 (deep purple, PDB ID 6LU7) from SARS-CoV-2. Upon inhibitor binding, residues Met49, Leu50, and Met165 change their conformations (curved black arrows), whereas the small helix with residues 46–50 and the β-hairpin loop with residues 166–170 move apart, resulting in the loop with residues 190–194 which accommodates the inhibitor's P5 substituent to shift closer to the β-hairpin loop. All distances are given in Ångstroms.

pH = 8, 150 mM NaCl, and 1 mM TCEP for 18 h at 4 °C to remove the C-terminal His$_6$ tag, resulting in a 3CL M$^{pro}$ with authentic N-termini and C-termini. The PreScission-treated 3CL M$^{pro}$ solution was applied to a HisTrap FF column to remove the PreScission protease, the C-terminal tag, and 3CL M$^{pro}$ with uncleaved His tag. The authentic 3CL M$^{pro}$ was collected in the flow-through and concentrated to 4 mg ml$^{-1}$.

**Crystallization**. The concentrated protein solution (4 mg ml$^{-1}$) was first sent to the High-Throughput Crystallization Screening Center at the Hauptman–Woodward Medical Research Institute (Buffalo, NY), where 1536 crystallization conditions were screened using 96-well sitting drop plates[28]. Thin plate-like crystal "flowers" appeared in several conditions within a week and were set up manually in the lab to reproduce the crystal growth. The best-looking crystals grew in 0.1 M BIS–TRIS pH = 6.5, 25% PEG3350. Several crystal "flower" aggregates were collected from this condition and were used to make microseeds using Hampton Research seed beads. For X-ray crystallography, crystals were grown in 10 µL drops made by mixing the protein sample (4 mg/mL), reservoir

solution (0.1 M BIS–TRIS pH = 6.5, 20% PEG3350) at a 1:1 ratio and 0.2 µL of microseeds (1:100 dilution) in a sitting drop setup. The resulting crystal drop pH was measured using a microelectrode to be 7.0. Single plate-like crystals grew in several days (Supplementary Fig. 1a). To collect a room-temperature diffraction dataset, a crystal of 3CL M$^{pro}$ was mounted using the MiTeGen (Ithaca, NY) room-temperature capillary setup (Supplementary Fig. 1b, c).

**X-ray data collection and structure refinement**. Room temperature X-ray crystallographic data for ligand-free 3CL M$^{pro}$ were collected on a Rigaku HighFlux HomeLab instrument equipped with a MicroMax-007 HF X-ray generator and Osmic VariMax optics. The diffraction images were obtained using an Eiger 4 M hybrid photon counting detector. Diffraction data were integrated using the CrysAlis Pro software suite (Rigaku Inc., The Woodlands, TX). Diffraction data were then reduced and scaled using the Aimless[29] program from the CCP4 suite[30], molecular replacement using PDB code 6M03 was then performed with Molrep[30] from the CCP4 program suite. Refinement of the protein structure was conducted using *Phenix.refine* from the Phenix[31] suite of programs and the COOT[27] molecular graphics program. The geometry of the final structure was then carefully checked with Molprobity[32]; the data collection and refinement statics are shown in Supplementary Table 2.

**MD simulation**. Classical MD simulation was prepared, conducted, and analyzed on an apo 3CL M$^{pro}$ dimer adapted from deposited structure PDB code 6Y84 using GROMACS 2020[33]. The system was described using the CHARMM36m force field[34]. The dimer was solvated in a rhombic dodecahedron with 10 Å from the nearest cell edge using the TIP3P[35] water model and 8 Na ions. Periodic boundary conditions were applied, and the system was minimized in less than 1000 steps using the steepest descent algorithm. The system was equilibrated to 300 K and 1 bar using the V-rescale thermostat[36] and Berendsen barostat[37]. A 1 µs production MD was performed using the leap-frog integration with Nose-Hoover[38,39] and Parinello–Rahman couplings[40]. All bonds involving a hydrogen atom were constrained using the SHAKE algorithm[41]. Atomic coordinates were saved every 10 ps. RMSF was calculated for protein backbone atoms after RMSD convergence. Reasonable invariance in radius of gyration ($R_g$) values of 26.2 ± 0.15 Å over the 1 µs timescale indicates compactness and stability of the protein dimer (Supplementary Fig. 3).

**Reporting summary**. Further information on research design is available in the Nature Research Reporting Summary linked to this article.

## Data availability

The structure and corresponding structure factors have been deposited into the Protein Data Bank with the PDB accession code 6WQF. Other data are available from the corresponding authors upon reasonable request.

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

## Acknowledgements

This work was supported by the Laboratory Directed Research and Development Program at Oak Ridge National Laboratory (ORNL). Research at ORNL's Spallation Neutron Source and HFIR was sponsored by the Scientific User Facilities Division, Office of Basic Energy Sciences, U.S. Department of Energy. The Office of Biological and Environmental Research supported research at ORNL's Center for Structural Molecular Biology (CSMB), using facilities supported by the Scientific User Facilities Division, Office of Basic Energy Sciences, U.S. Department of Energy. Funding for this project was provided in part by federal funds from the National Institute of Allergy and Infectious Diseases, National Institutes of Health, Department of Health and Human Services, under Contract HHSN272201700060C. Crystallization screening was supported through NSF grant 2029943. We thank Swati Pant, Kevin Weiss, Yichong Fan, Qiu Zhang of ORNL for their help with the plasmid preparation and initial protein expression.

## Author contributions

L.C., A.K., H.M.O'N., and P.L. designed the study. R.J., L.S., and A.J. designed and cloned the gene. D.W.K., H.M.O'N., and G.P. expressed and purified the protein. D.W.K. and A.K. crystallized the protein and collected the data. L.C. reduced the data and refined the structure. D.W.K. performed and analyzed molecular dynamics simulations. D.W.K., L.C., P.L., and A.K. wrote the paper with help from all co-authors.

## Competing interests

The authors declare no competing interests.
