## [Peer Review File · Nature Communications]

Reviewer #1 (Remarks to the Author):

The manuscript by Kneller and co-authors describes a room temperature crystal structure of a SARS-CoV-2 3CLMpro protease, which has been identified as one of the attractive targets for the design of antiviral inhibitors. Recently, complex structures of this protease have been reported, all determined at cryogenic temperatures using X-ray crystallography. It is known that room-temperature crystal structures are better suited to be used as receptors during in silico docking experiments. Therefore, the structure reported in this manuscript is an important contribution to support further development of antiviral inhibitors and drug repurposing and merits publication. I hope that this structure will be released very soon.

The PDB validation report confirms a good quality of the model and raises no questions.

I also have no further comments other than "Well done".

With kind regards,

Piotr Neumann

Reviewer #2 (Remarks to the Author):

The manuscript titled "Structural Plasticity of the SARS-CoV-2 3CL Mpro Active Site Cavity Revealed by Room Temperature X-ray Crystallography" addresses the room temperature X-ray crystal structure of unliganded SARS-CoV-2 3CL Mpro. The authors also compare this structure to the reported 3CL Mpro X-ray structure whose crystal structure was obtained at 100K. The authors made an excellent study showing the resting structure of the active site and the conformation of the catalytic site cavity at room temperature. That may represent the real structure occurring at the physiological condition than that of obtained at very low temperature and under stress.

It is original, nice flow and is well organized. I believe the manuscript is timely and will get the attention of many researchers studying computational modeling and drug design against COVID-19. The manuscript is publishable in the present form with the following comment:

Comment:

In the process of MD simulation, every step was carried out properly and nicely. It should be better if the authors provide graph of root-mean-squared deviation, root-mean-squared fluctuation, and radius of gyration (Rg) of the ligand unbound 6WQF structure (room temperature) and ligand unbound 6LU7 (100K) to show how the structures are changing dynamically over time. Especially, radius of gyration may provide an invaluable information about the compactness of the crystal structure obtained at room temperature.

Reviewer #3 (Remarks to the Author):

The work of Kneller et al describes the room temperature structure SARS-CoV-2 main protease. The crystallographic data and analysis are corresponding to the state-of-the-art. The quality of the data made it possible to verify the claims about altered conformation of the C-terminus and the changes in the vicinity to the active site. I am less than enthusiastic about the use of theory in the form of molecular dynamics simulation which appear to lack predictive and explanatory power and

at worst case leads to misleading interpretation. This is far from unique in the literature, but I think it is important to acknowledge the complete lack of insight this 1 μ s simulation provides. I doubt that the structure will improve docking predictions, not because of the experimental data quality, but because the docking algorithms do not take advantage of the data. The only novelty in the paper is the difference between structures at different temperatures where the protein atoms undergo different type of fluctuations. To consider this work seminal a genuine advancement of the theory is necessary. At the very least the discussion should decide if the theory was appropriate or not given its ability to make predictions and explanations. If it was not it should point to a promising new direction.

Main issue:

Figure S3 shows an RMSF plot shows magnitudes that are not compatible with having a stable structure even though the crystal structure appears to be well defined at least in some of these regions (for example C-terminus in Figure S2 looks like a single conformation). This happens even though the protein is "stapled together" with the SHAKE algorithm. What happens to the structure in an unconstrained simulation? If the simulated atoms in these loops are present at the experimentally determined locations only occasionally by chance, then I am afraid the simulation failed to fulfill its purpose and it should be rejected. A successful, unconstrained simulation should predict defined positions with periodic fluctuations around these positions to explain the experimental data.

Other comments:

Abstract and elsewhere

"...revealing the resting structure of the active site and the conformation of the catalytic site cavity."

What does "resting" mean? Why a low-temperature structure is not the resting structure? It is resting compared to the high-temperature denatured form? If anything, vibrational activity is larger at higher temperature and conformational changes can occur more readily. If a protein has a well-defined global structure it is forming and maintaining itself actively at moderately high temperatures. Protein folding has not been observed at cryogenic temperatures on the contrary there are many examples of proteins that become disordered at low temperatures. The active site with a different protonation state is just as "resting". If the protein is hydrolyzing substrates then we may see a transient structure, but none of the examples are such. Consider a more specific description for the state observed in the crystal structure.

"The 3CL Mpro structure is composed of three domains.^{18,19} Domains I (residues 8-101) and II (residues 102-184) are composed of antiparallel β -barrel structures and are the catalytic domains. Domain III (residues 201-303) is composed of five α -helices and is responsible for the enzyme dimerization. This helical domain plays an essential role in the protease function as the monomeric enzyme is not catalytically active. Thus, 3CL Mpro forms a functional dimer through intermolecular interactions, mainly between the helical domains (Figure1a)."

That the missing helical domain causes the protein to be inactive does not mean that the dimeric form is necessary for the activity. The explanation and reference which connects these different aspects is missing. (for example, Zhang et al Science, 2020)

Line 103 "We superimposed our room temperature structure of 3CL Mpro with one obtained at 100K (PDB)."

How was the superposition performed?

Minor issues:

Main text:

Line 81 (see Methods)

SI:

"was solvated in a rhombic dodecahedron with 10 \AA from the nearest cell edge using the TIP3P9 water model and 8 Na ions."

Unit missing

101 line "Completeness (%) 95.4 (72.999.2)"

Strange decimal notation.

What does no observation & $|F|=0$ means as data rejection criteria? Does it mean no rejection? It

is better to say this explicitly and it is only acceptable approach.

Line 118 "* Values in parentheses are for highest-resolution shell. Data were collected from 1 crystal for each structure."

Only one structure was reported, right?

We thank the three knowledgeable reviewers for helping us to improve our manuscript. Below please find a point by point discussion of how we addressed each reviewer's point. Reviewer comments appear in black text followed by our response in blue text.

Reviewer #1:

The manuscript by Kneller and co-authors describes a room temperature crystal structure of a SARS-CoV-2 3CLMpro protease, which has been identified as one of the attractive targets for the design of antiviral inhibitors. Recently, complex structures of this protease have been reported, all determined at cryogenic temperatures using X-ray crystallography. It is known that room-temperature crystal structures are better suited to be used as receptors during in silico docking experiments. Therefore, the structure reported in this manuscript is an important contribution to support further development of antiviral inhibitors and drug repurposing and merits publication. I hope that this structure will be released very soon.

The PDB validation report confirms a good quality of the model and raises no questions. I also have no further comments other than "Well done".

With kind regards,

Piotr Neumann

Response: We are grateful for this positive evaluation of our experiments and the manuscript. The structure has now been released to the worldwide scientific community by the PDB (<https://www.rcsb.org/structure/6WQF>).

Reviewer #2:

The manuscript titled "Structural Plasticity of the SARS-CoV-2 3CL Mpro Active Site Cavity Revealed by Room Temperature X-ray Crystallography" addresses the room temperature X-ray crystal structure of unliganded SARS-CoV-2 3CL Mpro. The authors also compare this structure to the reported 3CL Mpro X-ray structure whose crystal structure was obtained at 100K. The authors made an excellent study showing the resting structure of the active site and the conformation of the catalytic site cavity at room temperature. That may represent the real

structure occurring at the physiological condition than that of obtained at very low temperature and under stress.

It is original, nice flow and is well organized. I believe the manuscript is timely and will get the attention of many researchers studying computational modeling and drug design against COVID-19. The manuscript is publishable in the present form with the following comment:

Comment:

In the process of MD simulation, every step was carried out properly and nicely. It should be better if the authors provide graph of root-mean-squared deviation, root-mean-squared fluctuation, and radius of gyration (Rg) of the ligand unbound 6WQF structure (room temperature) and ligand unbound 6LU7 (100K) to show how the structures are changing dynamically over time. Especially, radius of gyration may provide an invaluable information about the compactness of the crystal structure obtained at room temperature.

Response: We would like to thank the reviewer for the positive assessment of our work. As requested, we have updated Figure S3 with an additional graph of root-mean-square deviation. Because we did not perform MD simulations on structure 6LU7, which contains a bound inhibitor, we did not provide the graphs for the inhibitor-bound complex. Besides, such simulations are run at 300K; therefore, differences in simulations for ligand-free structures are expected to be minimal.

Reviewer #3:

The work of Kneller et al describes the room temperature structure SARS-CoV-2 main protease. The crystallographic data and analysis are corresponding to the state-of-the-art. The quality of the data made it possible to verify the claims about altered conformation of the C-terminus and the changes in the vicinity to the active site. I am less than enthusiastic about the use of theory in the form of molecular dynamics simulation which appear to lack predictive and explanatory power and at worst case leads to misleading interpretation. This is far from unique in the literature, but I think it is important acknowledge the complete lack of insight this 1 μ s simulation provides. I doubt that the structure will improve docking predictions, not because of the experimental data quality, but because the docking algorithms do not take advantage of the data. The only novelty in the paper is the difference between structures at different temperatures where the protein atoms undergo different type of fluctuations. To consider this work seminal a genuine advancement of the theory is necessary. At the very least the discussion should decide if the theory was appropriate or not given its ability to make predictions and explanations. If it was not it should point to a promising new direction.

Response: We thank the reviewer for evaluating our manuscript. Our replies to the reviewer's specific questions are given below.

Main issue: Figure S3 shows an RMSF plot shows magnitudes that are not compatible with having a stable structure even though the crystal structure appears to be well defined at least in some of these regions (for example C-terminus in Figure S2 looks like a single conformation). This happens even though the protein is “stapled together” with the SHAKE algorithm. What happens to the structure in an unconstrained simulation? If the simulated atoms in these loops are present at the experimentally determined locations only occasionally by chance, then I am afraid the simulation failed to fulfill its purpose and it should be rejected. A successful, unconstrained simulation should predict defined positions with periodic fluctuations around these positions to explain the experimental data.

Response: We performed our MD simulations following standard protocols. Indeed, the simulation was run unconstrained, except for bonds to hydrogen atoms using the SHAKE algorithm. We have corrected the relevant sentence in the MD simulation portion of the Materials and Methods (Supporting Information) to read “All bonds involving a hydrogen atom were constrained using the SHAKE algorithm¹⁵.” Also, our MD simulation correctly predicts the flexibility of the structural regions that undergo significant change in conformation upon a ligand binding.

Other comments:

Abstract and elsewhere “...revealing the resting structure of the active site and the conformation of the catalytic site cavity.” What does “resting” mean? Why a low-temperature structure is not the resting structure? It is resting compared to the high-temperature denatured form? If anything, vibrational activity is larger at higher temperature and conformational changes can occur more readily. If a protein has a well-defined global structure it is forming and maintaining itself actively at moderately high temperatures. Protein folding has not been observed at cryogenic temperatures on the contrary there are many examples of proteins that become disordered at low temperatures. The active site with a different protonation state is just as “resting”. If the protein is hydrolyzing substrates then we may see a transient structure, but none of the examples are such. Consider a more specific description for the state observed in the crystal structure.

Response: A good point. We changed the “resting state” to “ligand-free state”.

“The 3CL Mpro structure is composed of three domains.^{18,19} Domains I (residues 8-101) and II (residues 102-184) are composed of antiparallel β -barrel structures and are the catalytic domains. Domain III (residues 201-303) is composed of five α -helices and is responsible for the enzyme dimerization. This helical domain plays an essential role in the protease function as the

monomeric enzyme is not catalytically active. Thus, 3CL Mpro forms a functional dimer through intermolecular interactions, mainly between the helical domains (Figure1a).” That the missing helical domain causes the protein to be inactive does not mean that the dimeric form is necessary for the activity. The explanation and reference which connects these different aspects is missing. (for example, Zhang et al Science, 2020)

Response: A useful point. We included relevant references (# 20-24).

Line 103 “We superimposed our room temperature structure of 3CL Mpro with one obtained at 100K (PDB..” How was the superposition performed?

Response: We used the standard least-squares superposition approach, as implemented in the COOT molecular graphics program. We have added this detail into the manuscript.

Minor issues: Main text:

Line 81 (see Methods) SI: “was solvated in a rhombic dodecahedron with 10 _ from the nearest cell edge using the TIP3P9 water model and 8 Na ions.” Unit missing

Response: In our Microsoft Word version of the document the Å unit is present. Perhaps, it disappears during conversion to the PDF file.

101 line ”Completeness (%) 95.4 (72.999.2)” Strange decimal notation. What does no observation & $|F|=0$ means as data rejection criteria? Does it mean no rejection? It is better to say this explicitly and it is only acceptable approach.

Response: We corrected the typographical error present in the completeness value. We have also removed the row for Data rejection criteria.

Line 118 “* Values in parentheses are for highest-resolution shell. Data were collected from 1 crystal for each structure.” Only one structure was reported, right?

Response: Correct, we have revised this sentence to read “* Values in parentheses are for highest-resolution shell. Data were collected from one crystal.”